

# Phonon dressing of a facilitated one-dimensional Rydberg lattice gas

**Matteo Magoni[1*], Paolo P. Mazza[1] and Igor Lesanovsky[1,2]**

**1** Institut für Theoretische Physik, Eberhard Karls Universität Tübingen,
Auf der Morgenstelle 14, 72076 Tübingen, Germany
**2** School of Physics and Astronomy and Centre for the Mathematics and Theoretical Physics
of Quantum Non-Equilibrium Systems, The University of Nottingham,
Nottingham, NG7 2RD, United Kingdom

* matteo.magoni@uni-tuebingen.de

## Abstract

We study the dynamics of a one-dimensional Rydberg lattice gas under facilitation (anti-blockade) conditions which implements a so-called kinetically constrained spin system. Here an atom can only be excited to a Rydberg state when one of its neighbors is already excited. Once two or more atoms are simultaneously excited mechanical forces emerge, which couple the internal electronic dynamics of this many-body system to external vibrational degrees of freedom in the lattice. This electron-phonon coupling results in a so-called phonon dressing of many-body states which in turn impacts on the facilitation dynamics. In our theoretical study we focus on a scenario in which all energy scales are sufficiently separated such that a perturbative treatment of the coupling between electronic and vibrational states is possible. This allows to analytically derive an effective Hamiltonian for the evolution of clusters of consecutive Rydberg excitations in the presence of phonon dressing. We analyze the spectrum of this Hamiltonian and show — by employing Fano resonance theory — that the interaction between Rydberg excitations and lattice vibrations leads to the emergence of slowly decaying bound states that inhibit fast relaxation of certain initial states.

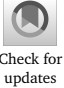

# 1   Introduction

In the past decades there has been a tremendous progress in the study of dynamical properties
of complex quantum many-body systems with cold atoms [1–3]. A significant role has been
played by Rydberg gases, in which atoms are excited to high-lying and strongly interacting
states [4–19]. Thanks to the strong state-dependent interactions between Rydberg excitations, Rydberg gases constitute an ideal experimental platform for the implementation and
simulation of so-called kinetically constrained quantum systems [20–23]. The phenomenology of such systems has been recently explored in several experiments involving bulk Rydberg
gas clouds [24] or reconfigurable optical tweezer arrays [25–27]. The results observed in
these experiments can be theoretically explained by the presence of a reduced connectivity
between different configurations in the Hilbert space [28–31]. Being first introduced for the
study of kinetic aspects in classical glassy systems [32], kinetically constrained systems have
been shown to possess peculiar dynamical properties [33–36], in relation to nucleation and
growth processes [37–39], the emergence of non-equilibrium phase transitions [40,41], localization [42–44] and the absence of relaxation and thermalization in general [45–48].

   In this work we analyze the influence of lattice vibrations on the dynamics of a kinetically
constrained one-dimensional Rydberg lattice gas. We focus on the so-called facilitation constraint, in which one Rydberg atom is favoured to (de)excite if only one neighboring Rydberg
atom is already excited [49–52]. Being held in harmonic traps, the atoms are subject to lattice vibrations which couple to Rydberg excitations. This results in a phonon dressing [53]
that affects the properties of the facilitation dynamics [54]. Throughout, we consider a parameter regime where the different energy scales involved in the problem are well separated.
This allows us to employ a perturbative expansion in terms of the coupling constant between
the Rydberg excitations (represented by effective spin degrees of freedom) and the phonon
modes. By integrating out the phonon degrees of freedom, we derive an effective Hamiltonian
describing the dynamics of phonon dressed clusters of consecutive Rydberg excitations. We
investigate its energy spectrum and study the dynamics of phonon dressed Rydberg clusters.
By using Fano resonance theory, we show that phonon dressing leads to a reduced mobility
of some cluster configurations which is caused by the emergence of bound states. This effect can be observed in the dynamics of the (Rydberg atom) density making it detectable in
experiments.

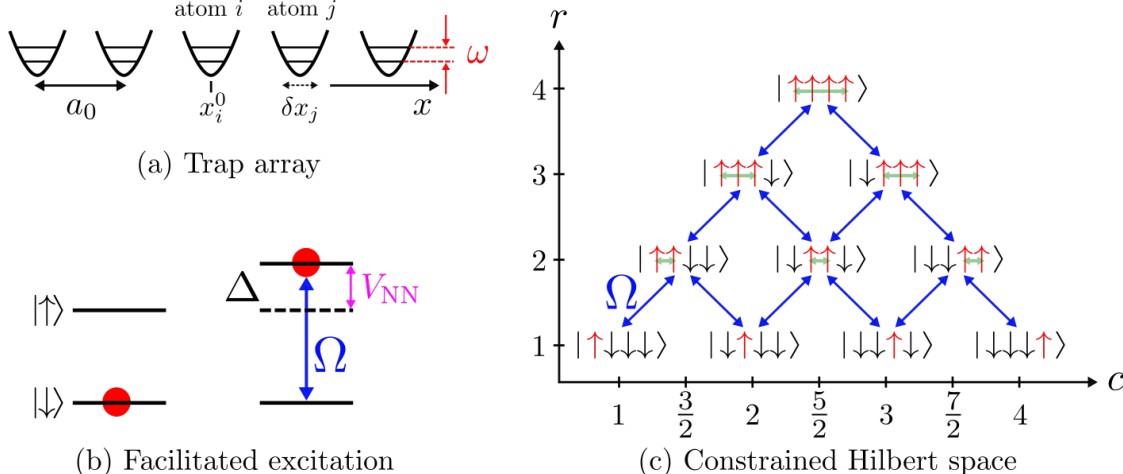

(a) Trap array

(b) Facilitated excitation

(c) Constrained Hilbert space

Figure 1: **Setting and structure of constrained Hilbert space**: (a) The system we consider consists of a one-dimensional lattice of $N$ harmonic traps with harmonic frequency $\omega$. The chain is orientated in $x$-direction and the spacing between adjacent traps is $a_0$. Each of the traps contains a single atom. The position of the center of the trap containing the $i$-th atom is denoted with $x_i^0$, while the displacement of the atom position from the respective trap center is $\delta x_i$. (b) Each atom is modeled as a two-level system, in which the states $|\downarrow\rangle$ and $|\uparrow\rangle$ represent the ground state and the (Rydberg) excited state, respectively. The atoms are excited with a laser with Rabi frequency $\Omega$ and detuning $\Delta$. The facilitation constraint is established when $\Delta + V_{\mathrm{NN}} = 0$, where $V_{\mathrm{NN}}$ denotes the interaction between two adjacent atoms in their respective equilibrium positions. (c) For $\Omega \ll \Delta$, a kinetically constrained dynamics is implemented, which takes place between resonant states. The constraint manifests in a reduced connectivity between states in the Hilbert space: starting from an initial single excitation, clusters of adjacent Rydberg excitations are formed. Such states are described in terms of two coordinates, $c$ and $r$, labeling the position of the center of mass and the number of excitations, respectively. Clusters containing at least two Rydberg excitations feature mechanical forces that act on the atoms on the edges of the excitation clusters (indicated by green arrows).

## 2 One-dimensional Rydberg lattice gas

We consider a one-dimensional chain of $N$ traps, separated by a nearest-neighbor distance $a_0$ and each being loaded with a single atom, as shown in Fig. 1. The electronic structure of each atom is described via a two-level system (effective spin $1/2$ particle), with the state $|\uparrow\rangle$ denoting the excited Rydberg state and the state $|\downarrow\rangle$ representing the ground state. Two atoms in the Rydberg state, located at sites $j$ and $k$, interact via a power-law potential $V(\boldsymbol{r}_j, \boldsymbol{r}_k) = V(|\boldsymbol{r}_j - \boldsymbol{r}_k|) = V(r_{j,k}) = C_\gamma r_{j,k}^{-\gamma}$. Here $\gamma = \{3, 6\}$, depending on the type of interaction (dipole-dipole or van der Waals) [3]. The Hamiltonian of the full system is given by

$$H = \sum_{j=1}^{N} \left( \frac{\Omega}{2} \hat{\sigma}_j^x + \Delta \hat{n}_j + \sum_{k<j} V(\boldsymbol{r}_j, \boldsymbol{r}_k) \hat{n}_j \hat{n}_k + \omega a_j^\dagger a_j \right), \tag{1}$$

where $\Omega$ is the Rydberg excitation laser Rabi frequency, $\hat{\sigma}^x = |\uparrow\rangle\langle\downarrow| + |\downarrow\rangle\langle\uparrow|$ is the spin flip operator, $\hat{n} = |\uparrow\rangle\langle\uparrow|$ projects onto the up state, $\Delta$ is the laser detuning from the atomic transition frequency and $\omega$ is the trap frequency. The operators $a_j^\dagger$ and $a_j$ are the phonon creation and annihilation operators at site $j$. These are defined with respect to the displacement of the

position $r_j$ of the $j$-th atom, from the center of the respective trap $r_j^0$: $\delta r_j = r_j - r_j^0$. Although in principle $\delta r_j$ is a vectorial quantity it is sufficient to consider only the phonon dynamics in $x$-direction, i.e. parallel to the chain. Then the fluctuations around the equilibrium positions are given in terms of bosonic operators as $\delta x_j = \sqrt{\frac{\hbar}{2m\omega}}(a_j^\dagger + a_j)$. This approximation relies on the fact that, if $|\delta r_j| \ll a_0$, which we assume throughout, the potential can be expanded around the equilibrium positions and approximated to leading order as

$$V(r_j, r_k) \simeq V(r_j^0, r_k^0) + \nabla V(r_j, r_k)\big|_{(r_j^0, r_k^0)} \cdot (\delta r_j, \delta r_k).$$

Since the interaction only depends on the relative distance between the atoms, $V(r_j, r_k) = C_\gamma r_{j,k}^{-\gamma}$, the gradient reads

$$\nabla V(r_j, r_k)\big|_{(r_j^0, r_k^0)} = -\frac{\gamma C_\gamma}{r_{j,k}^{\gamma+1}}(\hat{r}_{j,k}, -\hat{r}_{j,k})\bigg|_{(r_j^0, r_k^0)},$$

where $\hat{r}_{j,k} = \frac{r_j - r_k}{|r_j - r_k|}$ is the unit vector connecting the atom $k$ to the atom $j$. The gradient of the potential evaluated at $(r_j^0, r_k^0)$ has non-vanishing terms only in the $x$-components. Thus the only non zero component of the gradient is the one along the longitudinal direction. The expansion of the potential is then given by

$$V(r_j, r_k) \simeq V(r_j^0, r_k^0) - \frac{\gamma C_\gamma}{|x_j^0 - x_k^0|^{\gamma+1}}(\delta x_j - \delta x_k)$$

$$= V(r_j^0, r_k^0) - \frac{\gamma C_\gamma}{a_0^{\gamma+1}}\sqrt{\frac{\hbar}{2m\omega}}\left(a_j^\dagger + a_j - a_k^\dagger - a_k\right). \tag{2}$$

This expansion makes it evident that a simultaneous excitation of two atoms to the Rydberg state effectuates a coupling between the internal (electronic) and external (vibrational) degrees of freedom of the facilitated Rydberg chain.

## 3 Facilitated Rydberg dynamics

### 3.1 Hamiltonian of a single Rydberg cluster

We focus on the situation in which the dynamics of the system is subject to the facilitation constraint. This is achieved when the laser detuning $\Delta$ cancels out the interaction between two adjacent atoms, $V_{\rm NN} = V(r_j^0, r_{j+1}^0)$ in their respective equilibrium positions ($\Delta + V_{\rm NN} = 0$), as depicted in Fig. 1. Moreover, we assume that the next-nearest-neighbor interaction is small compared to the detuning, i.e. $V(r_j^0, r_{j+2}^0) \ll |\Delta|$, and that also the Rabi frequency of the laser is much smaller than the detuning $\Omega \ll |\Delta|$. These conditions lead to a constrained dynamics owed to the reduced connectivity between many-body states in the Hilbert space, which conserves the total number of clusters of consecutive Rydberg excitations in the lattice [55]. For example, when starting from a single excited Rydberg atom, the following states are connected (see also Fig. 1): $|\downarrow\uparrow\downarrow\downarrow \ldots\rangle \Leftrightarrow |\downarrow\uparrow\uparrow\downarrow\downarrow \ldots\rangle \Leftrightarrow |\downarrow\uparrow\uparrow\uparrow\downarrow \ldots\rangle \Leftrightarrow |\downarrow\downarrow\uparrow\uparrow\downarrow \ldots\rangle \Leftrightarrow |\downarrow\downarrow\uparrow\uparrow \ldots\rangle \Leftrightarrow \ldots$. This means that a cluster of consecutive excitations can expand or shrink, but cannot (dis)appear or split. When more than one cluster of consecutive Rydberg excitations is initially present, these clusters can also not merge.

Throughout this work we focus on a single cluster present in the lattice. In this case it is convenient to describe that state of a Rydberg cluster as a tensor product of its center of mass (CM) and relative coordinate

$$|\psi\rangle = |c\rangle \otimes |r\rangle. \tag{3}$$

Introducing these coordinates is particularly advantageous as they allow to reduce the complex many-body problem to a much simpler two-body problem, thanks to the kinetically constrained dynamics. Here $c$ labels the position of the CM of the cluster and $r$ denotes the number of excitations. In a lattice with $N$ sites with periodic boundary conditions, the CM coordinate can take $2N$ different values, $c = \frac{1}{2}, 1, \ldots, N$ (in units of the lattice spacing $a_0$), where half-integer and integer values refer to CM positions at the middle of a lattice spacing or at a lattice site respectively. The coordinate $r$ is an integer number between 1 and $N - 1$, since a cluster with $N$ excitations is not allowed. According to this notation, for instance, $|2\rangle |3\rangle = |\uparrow\uparrow\uparrow\downarrow\downarrow \ldots\rangle$ and $|\frac{5}{2}\rangle |2\rangle = |\downarrow\uparrow\uparrow\downarrow\downarrow \ldots\rangle$, as shown in Fig. 1c.

Given this representation, a state $|c\rangle |r\rangle$ is resonant with only four other states, provided that $1 < r < N - 1$ (when $r = 1$ the cluster can only increase, when $r = N - 1$ the cluster can only decrease). These are: $|c + \frac{1}{2}\rangle |r + 1\rangle$ (the spin to the right of the rightmost excitation flips up), $|c - \frac{1}{2}\rangle |r + 1\rangle$ (the spin to the left of the leftmost excitation flips up), $|c - \frac{1}{2}\rangle |r - 1\rangle$ (the rightmost excitation flips down), $|c + \frac{1}{2}\rangle |r - 1\rangle$ (the leftmost excitation flips down). Note, that the CM coordinate and the relative coordinate are not completely independent, as integer (half-integer) values of the CM position can be paired only with an odd (even) value for the relative coordinate. Such coupling between the relative and CM degrees of freedom of a cluster is a consequence of the discreteness of the lattice and does not appear in continuum space.

Using the expansion of the interaction potential, Eq. (2), and the representation in terms of the CM and relative coordinates, we can write the Hamiltonian of a single cluster of consecutive Rydberg excitations as

$$H = \Omega \sum_{c=\frac{1}{2}}^{N} \sum_{r=1}^{N-2} \left[ \left| c + \frac{1}{2} \right\rangle \langle c | \otimes (|r + 1\rangle \langle r | + \text{h.c.}) + \text{h.c.} \right] \tag{4}$$

$$- \kappa \sum_{c=\frac{1}{2}}^{N} \sum_{r=2}^{N-1} |c\rangle \langle c | \otimes |r\rangle \langle r | \left( a^\dagger_{c + \frac{r-1}{2}} + a_{c + \frac{r-1}{2}} - a^\dagger_{c - \frac{r-1}{2}} - a_{c - \frac{r-1}{2}} \right) + \omega \sum_{j=1}^{N} a^\dagger_j a_j .$$

The first term is the kinetic energy of the Rydberg cluster, while the second term contains the coupling between the degrees of freedom of the cluster and the phonons. The constant

$$\kappa = \sqrt{\frac{\hbar}{2m\omega}} \frac{\gamma C_\gamma}{a_0^{\gamma+1}} = \frac{x_0}{\sqrt{2}} \frac{\gamma C_\gamma}{a_0^{\gamma+1}} = \frac{\gamma}{\sqrt{2}} \frac{x_0}{a_0} V_{\text{NN}} , \tag{5}$$

quantifies the strength of this spin-phonon coupling. It depends on microscopic details, such as the gradient of the interaction potential (which for the power-law potential considered here can be expressed in terms of the nearest-neighbor interaction $V_{\text{NN}}$) and the harmonic oscillator length $x_0 = \sqrt{\hbar/(m\omega)}$. In case of a repulsive potential, that we consider in the following, $C_\gamma > 0$ and therefore $\kappa$ is a positive constant.

Note that, if a cluster is composed of $r$ consecutive excitations with the leftmost excitation at site $i_l$ and the rightmost one at site $i_r = i_l + r - 1$, then only the phonon operators corresponding to the harmonic traps on sites $i_l$ and $i_r$ couple to the cluster degrees of freedom. Indeed, the sum over all neighboring sites of Eq. (2) gives rise to a telescoping series of the phonon operators, whose sum is the difference between the operator corresponding to the position of the rightmost excitation and the one at the leftmost excitation of the cluster, whose position coordinates can be expressed in terms of $c$ and $r$.

## 3.2 Decoupling the relative and center of mass motion of a Rydberg cluster

In the next step we introduce phonon Fourier modes through $a_j = \frac{1}{\sqrt{N}} \sum_p A_p e^{ijp}$, with $p = \frac{2\pi}{N} k$ and $k = -\frac{N-1}{2}, \ldots, -1, 0, 1, \ldots, \frac{N-1}{2}$ (for odd $N$). Expressed in terms of the operators $A_p$, the

Hamiltonian reads

$$H = \Omega \sum_{c=\frac{1}{2}}^{N} \sum_{r=1}^{N-2} \left[ \left| c + \frac{1}{2} \right\rangle \langle c | \otimes (|r+1\rangle \langle r| + \text{h.c.}) + \text{h.c.} \right]$$
$$- \frac{\kappa}{\sqrt{N}} \sum_p \left[ 2i \sin \left( \frac{\hat{r}-1}{2} p \right) e^{i\hat{c}p} A_p + \text{h.c.} \right] + \omega \sum_p A_p^\dagger A_p \,, \tag{6}$$

where we have also introduced the operators $\hat{r} = \sum_{r=1}^{N-1} r |r\rangle \langle r|$ (the sum can start from $r=1$ thanks to the presence of the sine function) and $\hat{c} = \sum_{c=\frac{1}{2}}^{N} c |c\rangle \langle c|$.

The CM degree of freedom and the phonon modes can now be decoupled by applying the so-called Lee-Low-Pines (LLP) transformation [56] to Eq. (6), which is implemented through the unitary operator

$$U = \exp \left\{ -i\hat{c} \sum_p p A_p^\dagger A_p - i \frac{\pi}{2} \sum_p A_p^\dagger A_p \right\}.$$

By introducing the Fourier transform of the CM coordinate, $|c\rangle = \frac{1}{\sqrt{2N}} \sum_q e^{iqc} |q\rangle$, where $q = -2\pi + \frac{2\pi}{N} k$ with $k = 0, 1, \ldots, 2N-1$, the Hamiltonian can be finally be written in a block-diagonal form as $U^\dagger H U = \sum_q H_q |q\rangle \langle q|$. Hence, after the LLP and the Fourier transform, the label $q$ of the CM Fourier modes has become a good quantum number, and the Hamiltonian $H_q$ governing the evolution within a given $q$ sector is given by

$$H_q = 2\Omega \cos \left[ \frac{1}{2} \left( q + \sum_p p A_p^\dagger A_p \right) \right] \sum_{r=1}^{N-2} |r+1\rangle \langle r| + \text{h.c.}$$
$$- \frac{2\kappa}{\sqrt{N}} \sum_p \left[ \sin \left( \frac{\hat{r}-1}{2} p \right) \left( A_p + A_p^\dagger \right) \right] + \omega \sum_p A_p^\dagger A_p \,. \tag{7}$$

### 3.3 Effective Hamiltonian in the phonon dressing regime

In the following we will integrate or trace out the phonons, in order to obtain an effective phonon dressed facilitation dynamics of a Rydberg cluster. To this end we apply the unitary displacement operator

$$\hat{D} = \exp \left( \sum_p \hat{S}_p A_p^\dagger - \hat{S}_p A_p \right) \tag{8}$$

to Hamiltonian (7). Here

$$\hat{S}_p = \frac{2\kappa}{\omega \sqrt{N}} \sin \left( \frac{\hat{r}-1}{2} p \right) \tag{9}$$

is an hermitian operator that depends on the phonon momentum $p$. Under the application of the unitary $\hat{D}$, each phonon annihilation operator gets shifted as $\hat{D}^\dagger A_p \hat{D} = A_p + \hat{S}_p$. The displaced Hamiltonian $\tilde{H}_q = D^\dagger H_q D$ reads

$$\tilde{H}_q = \hat{D}^\dagger \left\{ 2\Omega \cos \left[ \frac{1}{2} \left( q + \sum_p p A_p^\dagger A_p \right) \right] \sum_{r=1}^{N-2} |r+1\rangle \langle r| \right\} \hat{D} + \text{h.c.} - \omega \sum_p \hat{S}_p^2 + \omega \sum_p A_p^\dagger A_p \,, \tag{10}$$

where $\hat{S}_p^2 = \frac{4\kappa^2}{\omega^2 N} \sin^2 \left( \frac{\hat{r}-1}{2} p \right)$ and $\sum_p \hat{S}_p^2 = 2 \frac{\kappa^2}{\omega^2} \sum_{r=2}^{N-1} |r\rangle \langle r|$. We did not explicitly evaluate here the displaced kinetic term. This is cumbersome, since $\hat{S}_p$ and $\sum_{r=1}^{N-2} |r+1\rangle \langle r|$ do not commute.

To make progress, nevertheless, we assume in the following that $\kappa \ll \omega$, i.e. that the interaction between the phonons and the Rydberg cluster dynamics is weak. We expand the

displaced kinetic term in powers of $\kappa/\omega$ and only retain terms up to order $(\kappa/\omega)^2$ (this is the same order as that of the term $\hat{S}_p^2$). To finally obtain the effective phonon dressed Hamiltonian, we project the displaced Hamiltonian onto the phonon vacuum, thus effectively tracing out the phonon degrees of freedom (see Appendix A for details). The effective "lattice-only" Hamiltonian for each CM mode $q$ then becomes

$$
\begin{aligned}
H_{\mathrm{eff},q} &= 2\Omega\left(1 - \frac{\kappa^2}{\omega^2}\right)\cos\left(\frac{q}{2}\right)\sum_{r=1}^{N-2}\left(|r+1\rangle\langle r| + |r\rangle\langle r+1|\right) - 2\frac{\kappa^2}{\omega}\sum_{r=2}^{N-1}|r\rangle\langle r| \\
&= J_q(\kappa)\hat{T} + \alpha(\kappa)|1\rangle\langle 1| - 2\frac{\kappa^2}{\omega},
\end{aligned}
\tag{11}
$$

where the last constant term will be neglected in the following. Here

$$
\hat{T} = \sum_{r=1}^{N-2}\left(|r+1\rangle\langle r| + |r\rangle\langle r+1|\right)
$$

is the kinetic energy (hopping) operator of the relative dynamics of the Rydberg cluster,

$$
J_q(\kappa) = 2\Omega\left(1 - \frac{\kappa^2}{\omega^2}\right)\cos\left(\frac{q}{2}\right)
\tag{12}
$$

is the renormalized hopping rate and

$$
\alpha(\kappa) = 2\frac{\kappa^2}{\omega}
\tag{13}
$$

is a "repulsive" potential shift acting on a cluster of length 1, i.e. containing only a single Rydberg atom. This potential shift reflects the peculiarity of such cluster, as it is the only one in which there are no Rydberg-Rydberg interactions. Consequently, since there are no mechanical forces, it is completely decoupled from the phonons.

In order to assess the quality of the performed approximations we compare in the following the band structure of the effective phonon dressed Hamiltonian

$$
H_{\mathrm{eff}} = \sum_q H_{\mathrm{eff},q}|q\rangle\langle q|,
\tag{14}
$$

with results from a numerical diagonalization of the full Hamiltonian (7). As can be seen in Fig. 2 the agreement is excellent for small values of $\kappa/\omega$, which is the regime where perturbation theory is expected to be valid. This suggests that the obtained effective model correctly describes the physics of phonon dressed Rydberg clusters. Moreover, the two bottom panels show that, for increasing strength of the phonon dressing, the uppermost energy level separates from the rest of the band. This separation can be explained by the emergence of a bound state, which is caused by the presence of the repulsive potential $\alpha(\kappa)$ [Eq. (5)] and which will be discussed in detail further below. Also visible is the narrowing of the bands due to the factor $1 - \kappa^2/\omega^2$ in the hopping rate, Eq. (12).

## 3.4 Experimental considerations

The perturbative expansion of the displacement operator in powers of $\kappa/\omega$ and the assumption of a coherent Rydberg cluster dynamics set certain constraints on the energy scales entering Hamiltonian (7) as well as the coherence time. In the following we will discuss whether these can be met in current experiments. Hamiltonian (7) is the sum of three terms, with $\Omega$, $\kappa$ and $\omega$ as the respective energy scales. A necessary condition for our perturbation theory to be

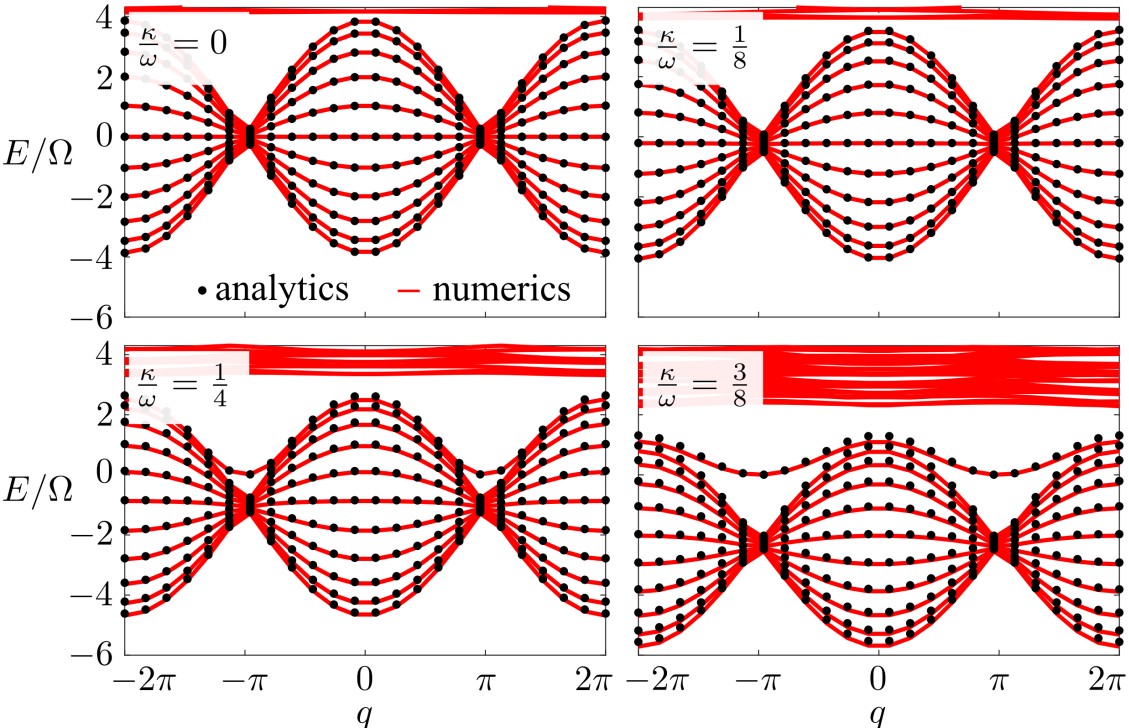

Figure 2: **Vibrationally dressed band structure of a single Rydberg cluster**: Energy bands in the free case ($\kappa = 0$) and with phonon dressing ($\kappa \neq 0$). Red lines are obtained through numerical diagonalization of Hamiltonian (7) with $N = 12$ sites and a truncation of the maximum number of phonons per site to 3. Black dots are the eigenvalues of the effective Hamiltonian (11) which has been obtained by integrating out the phonon degrees of freedom. The trap and Rabi frequencies are chosen such that $\omega = 8\,\Omega$. Note that, as $\kappa/\omega$ increases, the center of the band gets lower in energy. This is due to the presence of the constant term in Eq. (11) which is equal to $-2\kappa^2/\omega$ and is naturally included in the numerical diagonalization of Hamiltonian (7).

valid is that $\Omega, \kappa \ll \omega$, demanding that the trap frequency $\omega$ is much larger than the Rabi frequency $\Omega$ and the spin-phonon coupling constant $\kappa$. The trap frequency indeed measures the spacing between the zero-phonon band and the higher energy bands, while $\Omega$ determines the width of the zero-phonon band. The inequality $\Omega \ll \omega$ then ensures that the band with zero phonons remains well separated from the higher energy bands, avoiding undesired effects due to band mixing. The inequality involving $\kappa$ and $\omega$ is on the other hand necessary for the perturbative expansion to be valid. Both $\Omega$ and $\kappa$ are independent quantities, meaning that the derivation of the effective Hamiltonian (11) is rigorous in both situations where $\kappa$ is larger or smaller than $\Omega$. This is due to the fact that the displacement operator (8), that we expand perturbatively, depends on the ratio $\kappa/\omega$, but not on $\Omega$. Furthermore, in order to legitimately describe the coherent dynamics of phonon dressed Rydberg spin clusters with the effective Hamiltonian (11), the time scales involved therein must be considerably shorter than the Rydberg atom lifetime. Therefore — denoting with $\Gamma$ the decay rate of the Rydberg state to other atomic states — the perturbative expansion turns out to be valid once

$$\omega \gg \Omega, \kappa \gg \Gamma \tag{15}$$

is satisfied. However, the perturbation treatment is found to be surprisingly accurate even when some of these conditions are not strictly met: as shown in Fig. 2, where the trap and

Rabi frequencies are chosen such that $\omega = 8\,\Omega$, the agreement between the numerical diagonalization of the Hamiltonian (7) and the eigenvalues of the effective Hamiltonian (11) is excellent even though the zero-phonon band is close to the higher energy bands.

Next, we estimate the magnitude of the spin-phonon coupling constant, Eq. (5), for a system of $^{87}$Rb atoms. Assuming van der Waals interaction ($\gamma = 6$) among Rydberg atoms, this reduces to

$$\kappa = \frac{x_0}{\sqrt{2}} \frac{6\,C_6}{a_0^7}\,.$$

Choosing $a \simeq 5\,\mu m$ and $\omega \simeq 2\pi \times 300$ kHz, we obtain $x_0 \simeq 2 \times 10^{-2}\,\mu m$. The $C_6$ coefficient is proportional to $n^{11}$, where $n$ is the principal quantum number of the Rydberg state. For $n \simeq 60$ Rydberg $S$-state, $C_6 \simeq 140$ GHz $\mu m^6$ [57]. We therefore obtain the estimate

$$\kappa \simeq 2\pi \times 25\,\text{kHz}\,.$$

The lifetime for a Rydberg excitation with $n \simeq 60$ at $T = 300$ K is $\tau \simeq 10^{-4}s$. So the decay rate is $\Gamma \simeq 2\pi \times 1.6$ kHz [58], which is indeed significantly smaller than the spin-phonon coupling. Noting furthermore that a Rabi frequency of the Rydberg excitation laser on the order of $\Omega = \omega/8 \simeq 2\pi \times 37.5$ kHz is experimentally achievable [59], we see that the condition (15) can indeed be satisfied with the above parameter choices. The assumption $\Omega \ll |\Delta|$ necessary for the facilitation condition is also fulfilled because $|\Delta| = V_{\text{NN}} = C_6/a_0^6 \simeq 10$ MHz.

The most challenging condition is probably the assumption of a trap frequency of $\omega \simeq 2\pi \times 300$ kHz, which is larger than current typical values that are on the order of $\omega \simeq 2\pi \times 100$ kHz [60]. For this latter value one has $\kappa \simeq 2\pi \times 40$ kHz, making the ratio $\kappa/\omega = 0.4$, close to the case depicted in the bottom right of Fig. 2. In this case the Rabi frequency evaluates to $\Omega = \omega/8 \simeq 2\pi \times 12.5$ kHz, which reduces the ratio $\Omega/\Gamma$ to about 8 and therefore limits the time interval over which coherent evolution can be observed.

We assumed throughout that atoms in both their ground state and Rydberg state are trapped in the lattice potential. The feasibility of this has been demonstrated in Ref. [61], however, this is not yet standard technology in Rydberg quantum simulator setups. Furthermore, for the parameters considered, the spin-phonon coupling constant is about 15 times larger than the Rydberg atom decay rate. However, given that $\kappa$ depends on the gradient of the interaction potential, its value can be modified by tailoring the interaction potential between Rydberg states via microwave dressing, as theoretically discussed in Refs. [53, 62] and demonstrated in Ref. [63]. This may allow to push the ratio $\kappa/\omega$ in the region that is considered in Fig. 2.

We conclude this section by remarking that the parameter values discussed here represent the most ideal case in that they give rise to a scenario in which all energy scales are clearly separated. This is in fact very convenient for the theoretical analysis. In practice, it is reasonable to expect that also parameter choices that are less stringent will permit the experimental observation of signatures of phonon dressing in the dynamics of facilitated Rydberg clusters.

# 4 Dynamics of a phonon dressed Rydberg cluster

## 4.1 Numerical results

In this section we study the time evolution of a cluster initially prepared (at time $t = 0$) with a fixed CM position $c_0$ and a defined number of excitations $r_0$ as

$$|\psi(0)\rangle = |c_0\rangle \otimes |r_0\rangle\,.$$

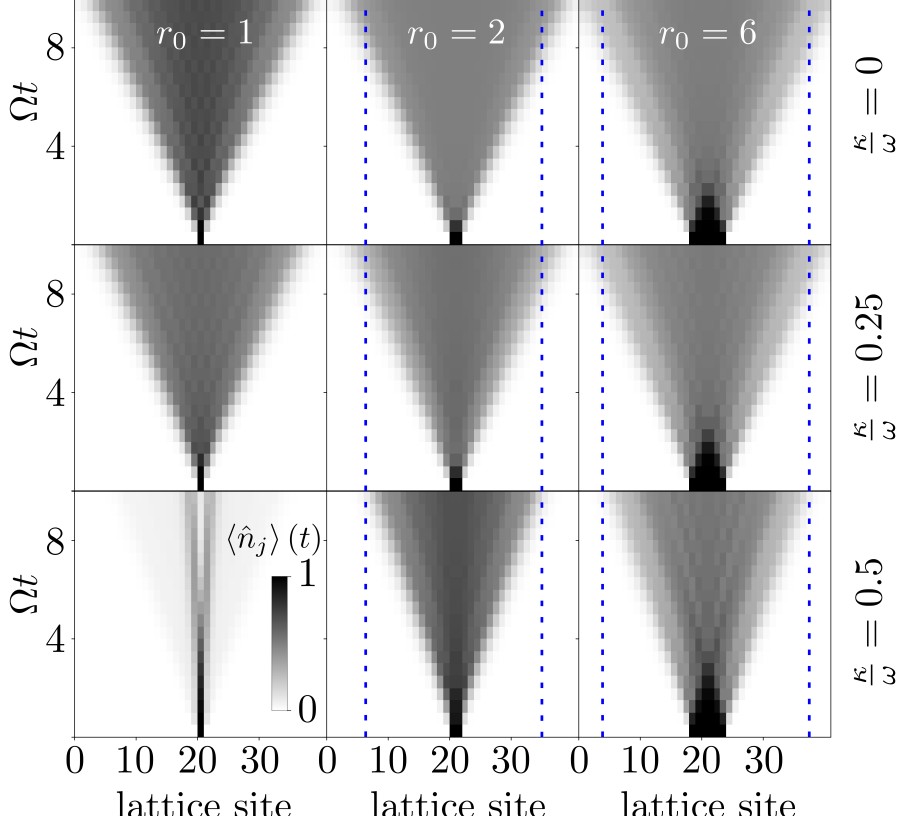

**Figure 3: Dynamics of a Rydberg cluster with $r_0$ consecutive initial excitations**:
Time evolution of the Rydberg excitation density $\langle \hat{n}_j \rangle(t)$ for different values of the
initial Rydberg cluster size, $r_0$, and spin-phonon coupling strength, $\kappa/\omega$. Visible
is a ballistic expansion, which becomes slower for large values of the spin-phonon
coupling constant $\kappa$. An almost complete inhibition of expansion appears in the case
$r_0 = 1$, as the strong repulsive potential makes transitions to propagating continuum
states off-resonant. The propagation of the Rydberg clusters with $r_0 > 1$ also slows
down with increasing spin-phonon interaction. This is due to the decrease of the
hopping rate $J_q$. The dotted blue lines are used to enhance the visibility of this effect.

This state evolves according to

$$|\psi(t)\rangle = e^{-iH_{\text{eff}}t} |\psi(0)\rangle = \frac{1}{\sqrt{2N}} \sum_q e^{iqc_0} |q\rangle \otimes e^{-iH_{\text{eff},q}t} |r_0\rangle \,, \qquad (16)$$

with each $q$ mode of the wave function evolving independently through the effective Hamiltonian (11).

Figure 3 shows the time evolution of the site-resolved Rydberg excitation density — a
quantity that can be experimentally measured [6] — for different values of $r_0$ and $\kappa/\omega$. For
$\kappa = 0$ (top three plots), the cluster undergoes ballistic expansion. This is indeed expected, as
in this case the effective Hamiltonian is simply given by the hopping term. As the ratio $\kappa/\omega$
increases, the value of the effective hopping rate $J_q(\kappa)$ becomes smaller, leading to a slowdown
of the ballistic expansion. The dashed blue lines, which are shown in the figure as a guide to the
eye, indicate this effect: the time needed for the cluster excitations to reach a given distance
from the initial location of the CM increases as the phonon dressing gets stronger. This effect is
more pronounced when the initial state has only one Rydberg excitation ($r_0 = 1$). The reason

for this is that this initial configuration is subjected to the repulsive potential $\alpha(\kappa)$, which is given by Eq. (11). This brings transitions from this initial state to other states off resonance and therefore inhibits relaxation, thereby yielding a rather pronounced manifestation of the phonon dressing.

# 5 Analytical results — Fano resonance theory

In the following we focus more closely on the scenario in which an initial state is prepared, that contains only a single excitation ($r_0 = 1$). This case, which corresponds to the left column in Fig. 3 is interesting, because it can to a large extent be analytically treated via Fano resonance theory [64]. This theory describes the interaction between a discrete state and a set of continuum states, and in the following we will show that our problem can be indeed mapped onto such situation. Exploiting this connection will allow to derive an analytical expression for the survival probability of a Rydberg cluster containing a single excitation, which yields further insights into the inhibition of relaxation observed in Fig. 3.

We start by rewriting the effective Hamiltonian (11) as

$$
\begin{aligned}
H_{\text{eff},q} &= J_q(\kappa) \sum_{r=2}^{N-2} (|r+1\rangle\langle r| + |r\rangle\langle r+1|) + \alpha(\kappa)|d\rangle\langle d| + J_q(\kappa)(|d\rangle\langle 2| + |2\rangle\langle d|) \\
&= \hat{H}_q^0 + \hat{V}_d + J_q(\kappa)(|d\rangle\langle 2| + |2\rangle\langle d|) .
\end{aligned}
\tag{17}
$$

Here, we use the state $|d\rangle$ to denote what we previously called state $|1\rangle$. It corresponds to the relative coordinate of a Rydberg cluster containing only a single excitation and will be identified as the discrete state in the framework of Fano theory. The energy of this state is $\alpha(\kappa)$ as given by Eq. (13) and the corresponding Hamiltonian is $\hat{V}_d$. This discrete state is coupled to one of the continuum states which interact through the Hamiltonian $\hat{H}_q^0$. The strength of this coupling $J_q(\kappa)$ is given by Eq. (12), which contains the dependence on the CM motion. For the sake of brevity we write in the following $J_q \equiv J_q(\kappa)$ and $\alpha \equiv \alpha(\kappa)$, leaving the dependence of these parameters on $\kappa$ implicit.

The Hamiltonian $\hat{H}_q^0$ is easily diagonalized and its eigenvalues $\{E_q^0(k)\}_{k=1,\dots,N-2}$ and normalized eigenvectors $|\bar{k}\rangle$, which satisfy $\hat{H}_q^0|\bar{k}\rangle = E_q^0(k)|\bar{k}\rangle$, are

$$
E_q^0(k) = 2J_q \cos\left(\frac{\pi}{N-1}k\right), \qquad k = 1,\dots,N-2
$$

and

$$
|\bar{k}\rangle = \sqrt{\frac{2}{N-1}} \sum_{r=2}^{N-1} \sin\left[\frac{\pi}{N-1}k(r-1)\right]|r\rangle .
\tag{18}
$$

Each eigenvector $|\bar{k}\rangle$ is therefore given as a superposition of the basis vectors $|r\rangle$ with which $\hat{H}_q^0$ was originally formulated [Eq. (17)]. We now proceed by choosing the vectors $\left\{|d\rangle, \{|\bar{k}\rangle\}_{k=1,\dots,N-2}\right\}$ as the new basis. With this change of basis, the Hamiltonian (17) is partially diagonalized, i.e. all continuum states are now mutually orthogonal. The analogy with the Fano resonance scenario becomes apparent by plotting the diagonal elements of the Hamiltonian (17), as shown in Fig. 4a: a discrete (bound) state, which represents a Rydberg cluster containing a single excitation, is coupled to a set of uncoupled continuum states. We also show for comparison the spectrum of the fully diagonalized Hamiltonian (17) in Fig. 4b: for $\alpha < |J_q|$, the spectrum is continuous and extends over the same range as the eigenenergies

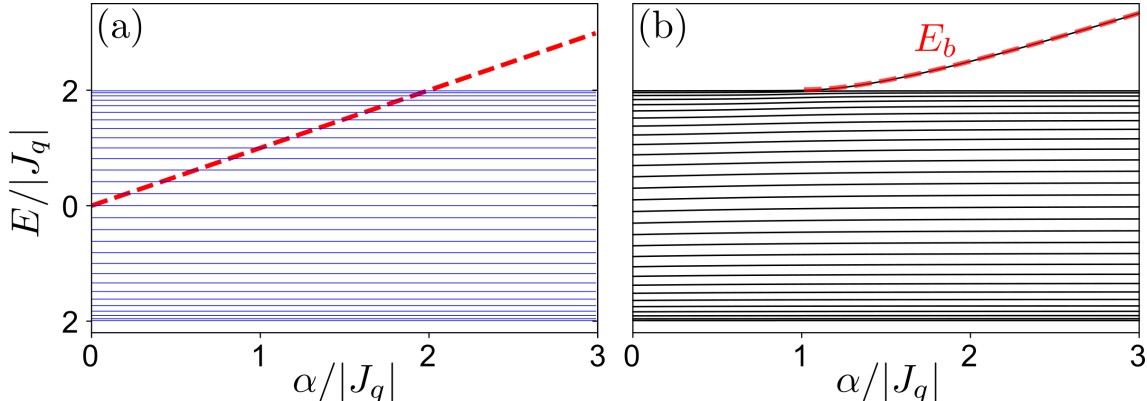

Figure 4: **Discrete state coupled to a continuum**: (a) Diabatic representation. Shown are the eigenvalues of $\hat{H}_q^0$ [see Eq. (17)] and the energy of the discrete state $|d\rangle$ (red dashed line). For $\alpha < 2|J_q|$, the discrete state is embedded inside the continuum. (b) Eigenvalues of the coupled Hamiltonian (17). When $\alpha > |J_q|$, a bound state possessing a large overlap with the state $|d\rangle$ emerges from the continuum. The red dashed line is the energy of the bound state, which is given by Eq. (19). Both panels are obtained with $N = 31$.

$E_q^0(k)$ of the uncoupled problem. For $\alpha > |J_q|$, a bound energy level with energy

$$E_b = \frac{\alpha^2 + J_q^2}{\alpha} \tag{19}$$

emerges (see derivation in Appendix B), which separates from the continuum band as $\alpha$ is increased. This bound state possesses a large overlap with the state $|d\rangle$. As shown below, the existence of such a bound state and the consequent modification of the spectrum of Hamiltonian (17) as a function of $\alpha/|J_q|$ are responsible for the strong inhibition of the expansion of a Rydberg cluster containing a single excitation (bottom left panel in Fig. 3).

Such cluster is represented by the state $|\psi(0)\rangle = |c_0\rangle \otimes |d\rangle$. Here $c_0$ denotes the initial CM position, which has to assume an integer number because it is paired with an odd value for the relative coordinate (Rydberg cluster of length 1, represented by $|d\rangle$), as discussed below Eq. (3). Each of the Fourier $q$ modes contributing to the CM state $|c_0\rangle$ evolves under the effective Hamiltonian (17) according to Eq. (16).

In the following we compute the (survival) probability $p_d(t)$ for each Fourier component, i.e. the probability for the system to remain in the initial state $|d\rangle$ at time $t$. To start, we explicitly write the matrix elements of Hamiltonian (17) in the new basis $\left\{ |d\rangle, \{|\bar{k}\rangle\}_{k=1,\ldots,N-2} \right\}$:

$$\begin{cases} \langle d|H_{\text{eff},q}|d\rangle = \alpha, \\ \langle d|H_{\text{eff},q}|\bar{k}\rangle = V(k), \\ \langle \bar{k}|H_{\text{eff},q}|\bar{k}'\rangle = E_q^0(k)\,\delta_{k,k'}, \end{cases} \tag{20}$$

with the real valued function

$$V(k) = J_q \sqrt{\frac{2}{N-1}} \sin\left(\frac{\pi}{N-1}k\right) \tag{21}$$

describing the coupling between the discrete state and the continuum. A generic eigenstate of Hamiltonian (17) can be written as

$$|\psi_E\rangle = a(E)|d\rangle + \sum_{k=1}^{N-2} b_k(E)|\bar{k}\rangle, \tag{22}$$

where the amplitudes $a$ and $b_k$ depend on the corresponding eigenvalue $E$. Each eigenvalue $E$ of course depends on $q$, but this dependence is left implicit in the notation for the sake of brevity. In order to obtain an expression for the survival probability, the key quantity to determine is the amplitude $a(E)$. This is because, according to Eq. (22), the survival probability is given by

$$p_d(t) = \left|\langle d|e^{-iH_{\text{eff},q}t}|d\rangle\right|^2 = \left|\sum_E |a(E)|^2 e^{-iEt}\right|^2 . \tag{23}$$

Here, the sum runs over the eigenvalues of the coupled Hamiltonian (17), which actually are the energy levels shown in Fig. 4b. This sum hence contains the contribution coming from the energies in the continuum, but, when $\alpha > |J_q|$, also the bound state with energy $E_b$ must be considered.

After some calculation detailed in Appendix B, one finds that the general expression for the survival probability is

$$p_d(t) = \left|\frac{\alpha^2 - J_q^2}{\alpha^2}\, e^{-iE_b t}\, \Theta(\alpha^2 - J_q^2) + \frac{2J_q^2}{\pi(\alpha^2 + J_q^2)} \int_0^\pi dx\, \frac{\sin^2 x}{1 - \frac{2\alpha J_q}{\alpha^2 + J_q^2}\cos x}\, e^{-i2J_q t\cos x}\right|^2 , \tag{24}$$

where $\Theta$ is the Heaviside step function. This exact result is the squared of a sum of two terms. The second one is the contribution to the survival probability stemming from the coupling of the discrete state to the continuum. It involves an integration, which is convergent since $\left|2\alpha J_q/\left(\alpha^2 + J_q^2\right)\right| \le 1$ for any value of $\alpha$ and $J_q$ (the integral can also be expressed by a convergent series of Bessel functions). The first term appears only for Fourier modes for which $\alpha > |J_q|$, and depends on time only through a phase which involves the bound state energy $E_b$.

For sufficiently long times the integral in the second term vanishes, and hence the survival probability at late times is approximately given by $\left|(\alpha^2 - J_q^2)/\alpha^2\right|^2$. This value tends to 1 as the ratio $\alpha/|J_q|$ increases. This explains the restricted mobility of the single excitation cluster shown in the bottom left corner of Fig. 3. Indeed, as $\alpha$ gets larger, there are more modes $q$ for which the condition $\alpha > |J_q|$ is satisfied, leading to a overall larger survival probability $p_d$ at late times. This is explicitly illustrated in Fig. 5, where the survival probability obtained from the numerical evaluation of Eq. (23) is compared with the analytical result (24). The three panels are organized such that the spin-phonon coupling constant increases from left to right, while the considered three modes $q$ are kept fixed. In the non-interacting case ($\alpha = 0$), the survival probability associated to all the modes $q$ decays to 0 accordingly to Eq. (B.10) given in Appendix B. For increasing value of $\alpha$, for more and more Fourier modes the inequality $\alpha > |J_q|$ is satisfied and the number of modes $q$ for which $p_d$ reaches a plateau at long times increases. This explains the inhibition of relaxation observed for a Rydberg cluster containing a single excitation.

## 6 Conclusion

We have considered a one-dimensional Rydberg lattice gas under facilitation conditions, which mimics the features of a kinetically constrained spin model. We have shown how the coupling between electronic and vibrational degrees of freedom — which is caused by the emergence of state-dependent forces — impacts on the dynamics of Rydberg excitations. This dressing of Rydberg excitations by phonons manifests in a reduction of the velocity with which facilitated clusters of consecutive Rydberg atoms grow over time. This becomes particularly apparent for

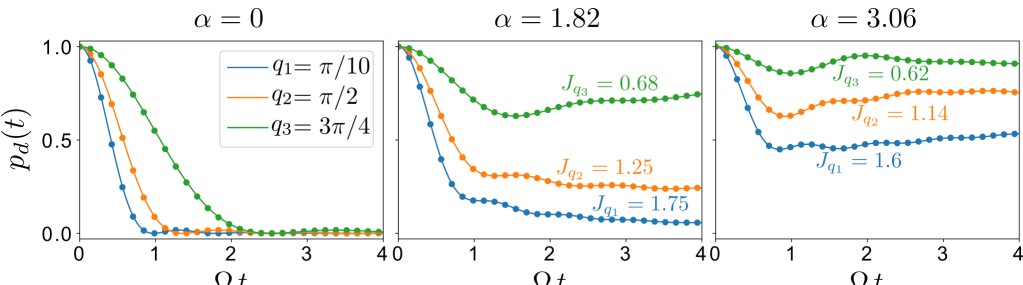

Figure 5: **Survival probability of a Rydberg cluster containing a single excitation**: The survival probability obtained from the numerical evaluation of Eq. (23) (plotted with dots) is exactly reproduced by the analytical result, Eq. (24), obtained from Fano theory and plotted with full lines. For $\alpha = 0$ the survival probability decays quickly to zero for all the three considered $q$ values. As $\alpha$ increases, more $q$-modes acquire a non zero survival probability. This explains the strong inhibition of the spreading of a Rydberg cluster containing a single excitation, as observed in Fig. 3. The parameters chosen for the plots are $\omega = 8\Omega$ and $\kappa = \{0, 2.7\Omega, 3.5\Omega\}$.

clusters that initially contain only a single Rydberg excitation. Using a perturbative approach in the strength of the spin-phonon coupling constant, we obtain an effective Hamiltonian for the dynamics of dressed Rydberg excitations, which accurately reproduces the band structure of the full system. Using an approach inspired by Fano resonance theory, we analytically derive an exact expression for the survival probability of the Rydberg cluster containing a single excitation, providing an explanation for the observed inhibition of relaxation.

Signatures of the reported dynamical features should be observable on current quantum simulator platforms based on atomic arrays [6]. However, reaching a regime in which all energy scales are separated in a way which we exploited for our analytical calculations may be challenging. Nevertheless, basic features, such as an impact of the lattice vibration on the propagation of excitations are expected to manifest also in settings that are currently accessible. In the future it would be interesting to consider phonon dressing of Rydberg excitations in high-dimensional lattices. Here, the physics is expected to be significantly richer: for example, the interaction between electronic and vibrational degrees of freedom will depend on the shape of the Rydberg clusters. It would, moreover, be interesting to study situations in which clusters interact or scatter off one another [65, 66].

# Acknowledgements

We are grateful for discussions with C. Groß, R. Eberhard, L. Steinert and P. Osterholz.

**Funding information** We acknowledge support from the "Wissenschaftler Rückkehr-programm GSO/CZS" of the Carl-Zeiss-Stiftung and the German Scholars Organization e.V., as well as by the Deutsche Forschungsgemeinschaft through SPP 1929 (GiRyd), Grant No. 428276754.

# A    Derivation of $H_{\text{eff},q}$

Here we derive the effective displaced Hamiltonian given by Eq. (11) in the main text. The derivation requires the following steps. First, we expand the displaced hopping term in Eq. (10)

in powers of $\kappa/\omega$. Secondly, we only keep the terms of the expansion up to order $(\kappa/\omega)^2$, to be consistent with the operator $\hat{S}_p^2$ which is also of order $(\kappa/\omega)^2$. Finally, the displaced Hamiltonian is projected onto the phonon vacuum state, which amounts to "integrating out" the phonons.

Let us rewrite the expression of the displaced hopping term present in Eq. (10) in the main text:

$$H_{\text{hop}} = \hat{D}^\dagger \left\{ 2\Omega \cos\left[ \frac{1}{2}\left( q + \sum_p p A_p^\dagger A_p \right) \right] \right\} \hat{D} \, \hat{D}^\dagger \left\{ \sum_{r=1}^{N-2} |r+1\rangle \langle r| \right\} \hat{D} + \text{h.c.} \,, \tag{A.1}$$

which, thanks to the identity $\hat{D}\hat{D}^\dagger = \mathbb{1}$, is given by a product of two displaced operators.

Now we proceed by computing the two factors separately. Since $\hat{D} = \prod_p e^{\hat{S}_p(A_p^\dagger - A_p)}$, the second displaced operator can be computed as

$$D^\dagger \left\{ \sum_{r=1}^{N-2} |r+1\rangle \langle r| \right\} D = \prod_p e^{-\hat{S}_p(A_p^\dagger - A_p)} \left( \sum_{r=1}^{N-2} |r+1\rangle \langle r| \right) \prod_q e^{\hat{S}_q(A_q^\dagger - A_q)}$$

$$= \sum_{r=1}^{N-2} \prod_p e^{-\frac{2\kappa}{\omega\sqrt{N}}\sin\left(\frac{r}{2}p\right)(A_p^\dagger - A_p)} \prod_q e^{\frac{2\kappa}{\omega\sqrt{N}}\sin\left(\frac{r-1}{2}q\right)(A_q^\dagger - A_q)} |r+1\rangle \langle r|$$

$$= \sum_{r=1}^{N-2} \prod_p \left( e^{-\frac{2\kappa}{\omega\sqrt{N}}\sin\left(\frac{r}{2}p\right)(A_p^\dagger - A_p)} e^{\frac{2\kappa}{\omega\sqrt{N}}\sin\left(\frac{r-1}{2}p\right)(A_p^\dagger - A_p)} \right) |r+1\rangle \langle r|$$

$$= \sum_{r=1}^{N-2} \prod_p e^{-\frac{2\kappa}{\omega\sqrt{N}}\left[\sin\left(\frac{r}{2}p\right) - \sin\left(\frac{r-1}{2}p\right)\right](A_p^\dagger - A_p)} |r+1\rangle \langle r|$$

$$= \sum_{r=1}^{N-2} e^{\sum_p -\frac{2\kappa}{\omega\sqrt{N}}\left[\sin\left(\frac{r}{2}p\right) - \sin\left(\frac{r-1}{2}p\right)\right](A_p^\dagger - A_p)} |r+1\rangle \langle r| \,,$$

where from the 3$^{\text{rd}}$ to the 4$^{\text{th}}$ row we make use of the property of the displacement operators $D(\alpha)D(\beta) = e^{(\alpha\beta^* - \alpha^*\beta)/2}D(\alpha + \beta)$, where in our case $\alpha = \beta = \hat{S}_p = \hat{S}_p^\dagger$. For $\kappa \ll \omega$, the exponential can be expanded in powers of $\kappa/\omega$ and the previous expression can be approximated as

$$D^\dagger \left\{ \sum_{r=1}^{N-2} |r+1\rangle \langle r| \right\} D \simeq$$

$$\simeq \sum_{r=1}^{N-2} |r+1\rangle \langle r| + \sum_{r=1}^{N-2} \sum_p \left( A_p - A_p^\dagger \right) \left[ S_p(r+1) - S_p(r) \right] |r+1\rangle \langle r| \tag{A.2}$$

$$+ \frac{1}{2!} \sum_{r=1}^{N-2} \sum_p \sum_v \left( A_p - A_p^\dagger \right) \left[ S_p(r+1) - S_p(r) \right] \left( A_v - A_v^\dagger \right) \left[ S_v(r+1) - S_v(r) \right] |r+1\rangle \langle r| \,,$$

where $S_p(r) = \frac{2\kappa}{\omega\sqrt{N}} \sin\left( \frac{r-1}{2}p \right)$ is the eigenvalue of the operator $\hat{S}_p$ relative to the eigenstate $|r\rangle$.

Now let us focus on the first displaced operator in Eq. (A.1). We write

$$\hat{D}^\dagger \left\{ 2\Omega \cos\left[ \frac{1}{2}\left( q + \sum_p p A_p^\dagger A_p \right) \right] \right\} \hat{D} = e^X Y e^{-X} \,,$$

where

$$X = \sum_p \hat{S}_p A_p - \hat{S}_p A_p^\dagger \quad \text{and} \quad Y = 2\Omega \cos\left[ \frac{1}{2}\left( q + \sum_p p A_p^\dagger A_p \right) \right] \,.$$

Using the Baker-Campbell-Hausdorff formula and truncating it at second order yields

$$e^X Y e^{-X} = Y + [X,Y] + \frac{1}{2!}[X,[X,Y]] + \dots$$

$$\simeq Y + (XY - YX) + \frac{1}{2!}(XXY + YXX - 2XYX). \tag{A.3}$$

The idea now is to gather Eq. (A.3) and Eq. (A.2) to collect the terms of orders $\kappa/\omega$ and $(\kappa/\omega)^2$. We then project these terms on the subspace with no phonons by computing the braket $\langle 0_{\mathrm{ph}}|\text{Eq. (A.3)} \cdot \text{Eq. (A.2)}|0_{\mathrm{ph}}\rangle$. All the terms of order $\kappa/\omega$ are proportional to $\langle 0_{\mathrm{ph}}|A_p|0_{\mathrm{ph}}\rangle$ and $\langle 0_{\mathrm{ph}}|A_p^\dagger|0_{\mathrm{ph}}\rangle$ and therefore they vanish. The matrix element evaluated for the terms of order $(\kappa/\omega)^2$ is instead non zero and, using the relation $\hat{S}_p \sum_{r=1}^{N-2}|r+1\rangle\langle r| = \sum_{r=1}^{N-2} S_p(r+1)|r+1\rangle\langle r|$, is given by

$$\langle 0_{\mathrm{ph}}|\text{Eq. (A.3)} \cdot \text{Eq. (A.2)}|0_{\mathrm{ph}}\rangle =$$

$$= \Omega \sum_{r=1}^{N-2}\sum_{p}\left\{2\cos\left(\frac{q+p}{2}\right)S_p(r+1)S_p(r) - \cos\frac{q}{2}\left[S_p^2(r+1)+S_p^2(r)\right]\right\}|r+1\rangle\langle r|.$$

By computing explicitly the sums over $p$ one obtains that $\sum_p \cos\left(\frac{q+p}{2}\right)S_p(r+1)S_p(r) = \cos\frac{q}{2}\frac{\kappa^2}{\omega^2}$ for $r > 1$ (if $r = 1$ it is equal to 0), $\sum_p S_p^2(r) = 2\frac{\kappa^2}{\omega^2}$ for $r > 1$ (if $r = 1$ it is equal to 0) and $\sum_p S_p^2(r+1) = 2\frac{\kappa^2}{\omega^2}$ $\forall r$. This braket can thus be rewritten as $-2\Omega\frac{\kappa^2}{\omega^2}\cos\frac{q}{2}\sum_{r=1}^{N-2}|r+1\rangle\langle r|$. Taking also into account the zeroth order, $(\kappa/\omega)^0$, the displaced hopping term Eq. (A.1) finally reduces to

$$H_{\mathrm{hop}} = 2\Omega\left(1 - \frac{\kappa^2}{\omega^2}\right)\cos\frac{q}{2}\sum_{r=1}^{N-2}|r+1\rangle\langle r|, \tag{A.4}$$

which is the first term of Eq. (11) in the main text.

## B  Derivation of the survival probability $p_d(t)$

We derive here the expression for the survival probability $p(t)$ given by Eq. (24). The derivation involves a sequence of steps which are detailed in the following: first, we derive the eigenvalue equation for $H_{\mathrm{eff},q}$ and obtain the expression of the bound state energy $E_b$. Then we calculate the general expression of $|a(E)|^2$ appearing in Eq. (23). This allows us to compute finally the survival probability $p_d(t)$.

Inserting Eq. (22) in the Schrödinger equation $H_{\mathrm{eff},q}|\psi_E\rangle = E|\psi_E\rangle$ and using Eq. (20), one obtains a system of equations in the unknowns $a = a(E)$ and $b_k = b_k(E)$ (the dependence on $E$ will be indicated explicitly only where necessary):

$$\begin{cases} \alpha a + \sum_{k=1}^{N-2} V(k) b_k = E a, \\ V(k) a + E_q^0(k) b_k = E b_k, \end{cases} \tag{B.1}$$

where $V(k)$ is the interaction potential given by Eq. (21) of the main text. As shown in Fig. 4b, the eigenvalues $E$, except for the bound state energy $E_b$, extend over the same range to which the uncoupled energies $E_q^0(k)$ belong. For large $N$, the uncoupled energies $E_q^0(k)$ form a continuous band and the eigenvalues $E$ included in this range degenerate to the energies $E_q^0(k)$. Therefore, in order to account for the occurrence of $E = E_q^0(k)$, the formal solution of the

second equation reads [67]

$$b_k = \left[ \frac{1}{E - E_q^0(k)} + z(E)\,\delta(E - E_q^0(k)) \right] V(k)\,a, \tag{B.2}$$

with the understanding that, when summed over $k$, one has to take the principal value (P.V.) of the sum over $(E - E_q^0(k))^{-1}$. The function $z(E)$ depends on energy, and for scattering problems one usually has conditions that imply $z = i\pi$ [68]. Here, instead, $z(E)$ is real and is determined by substituting the expression of $b_k$ in the first equation of (B.1). After factoring out the coefficient $a$, this gives

$$\alpha + \text{P.V.} \sum_{k=1}^{N-2} \frac{V^2(k)}{E - E_q^0(k)} + z(E) \sum_{k=1}^{N-2} V^2(k)\,\delta(E - E_q^0(k)) = E. \tag{B.3}$$

This is the eigenvalue equation whose solutions $E$ are the $N-1$ eigenvalues of $H_{\text{eff},q}$. By explicitly computing the two sums, one obtains the expression for $z(E)$. By noticing from Eq. (21) that

$$V^2(k) = \frac{4J_q^2 - E_q^0(k)^2}{2(N-1)}, \tag{B.4}$$

which expresses the interaction potential as a function of the energy in the continuum, the first sum can be computed as

$$\text{P.V.} \sum_{k=1}^{N-2} \frac{V^2(k)}{E - E_q^0(k)} = \frac{1}{2(N-1)} \text{P.V.} \sum_{k=1}^{N-2} \frac{4J_q^2 - E^2 + E^2 - E_q^0(k)^2}{E - E_q^0(k)}$$

$$= \frac{1}{2(N-1)} \left[ (N-2)E + (4J_q^2 - E^2)\text{P.V.} \sum_{k=1}^{N-2} \frac{1}{E - 2J_q \cos\left(\frac{\pi}{N-1}k\right)} \right].$$

The last principal value can be computed using

$$\text{P.V.} \sum_{k=1}^{N-2} \frac{1}{E - 2J_q \cos\left(\frac{\pi}{N-1}k\right)} \simeq \text{P.V.} \int_1^{N-2} \frac{1}{E - 2J_q \cos\left(\frac{\pi}{N-1}k\right)} dk$$

$$= \frac{N-1}{\pi} \text{P.V.} \int_{\frac{\pi}{N-1}}^{\frac{\pi(N-2)}{N-1}} \frac{1}{E - 2J_q \cos x} dx$$

$$= \frac{2(N-1)}{\pi} \text{P.V.} \int_{\tan\frac{\pi}{2(N-1)}}^{\tan\frac{\pi(N-2)}{2(N-1)}} \frac{1}{E - 2J_q + (E + 2J_q)t^2} dt$$

$$= \begin{cases} \frac{N-1}{\sqrt{E^2 - 4J_q^2}} & \text{if } E^2 - 4J_q^2 > 0, \\ 0 & \text{if } E^2 - 4J_q^2 < 0, \end{cases}$$

where we have taken the large $N$ limit and used the following substitutions:

$$x = \frac{\pi}{N-1}k, \qquad \cos x = \frac{1-t^2}{1+t^2}, \quad t = \tan\frac{x}{2}.$$

The second sum in Eq. (B.3) gives

$$\sum_{k=1}^{N-2} V^2(k)\,\delta(E - E_q^0(k)) = V^2(E)\,\rho(E)\,\Theta(4J_q^2 - E^2)$$

$$= \frac{\sqrt{4J_q^2 - E^2}}{2\pi}\,\Theta(4J_q^2 - E^2),$$

where we have used Eq. (B.4) and defined

$$\rho(E) = \left| \frac{dk}{dE_q^0(k)} \right|_{k=(E_q^0)^{-1}(E)} = \frac{N-1}{\pi\sqrt{4J_q^2 - E^2}}$$

as the density of states of the continuum $\{|\bar{k}\rangle\}$. Collecting all the terms and taking the large $N$ limit, the eigenvalue equation Eq. (B.3) now reads

$$\alpha + \frac{E}{2} - \frac{\sqrt{E^2 - 4J_q^2}}{2} \Theta(E^2 - 4J_q^2) + z(E) \frac{\sqrt{4J_q^2 - E^2}}{2\pi} \Theta(4J_q^2 - E^2) = E, \tag{B.5}$$

where $\Theta$ is the Heaviside step function. The energy of the bound state $E_b$, satisfying $E^2 - 4J_q^2 > 0$ and appearing only when $\alpha > |J_q|$, can be obtained by Eq. (B.5) as

$$E_b = \frac{\alpha^2 + J_q^2}{\alpha}. \tag{B.6}$$

It is plotted in Fig. 4b of the main text. Eq. (B.5) also provides the expression for the function $z(E)$

$$z(E)\Theta(4J_q^2 - E^2) = \pi \frac{E - 2\alpha + \sqrt{E^2 - 4J_q^2}\,\Theta(E^2 - 4J_q^2)}{\sqrt{4J_q^2 - E^2}}, \tag{B.7}$$

that is well defined only for $4J_q^2 - E^2 > 0$, i.e. when the eigenvalue $E$ is in the continuum. By enforcing the normalization condition

$$\langle \psi_E | \psi_{E'} \rangle = \delta_{E,E'},$$

using Eqs. (B.2), (B.3) as well as the properties of the Dirac delta distribution and the principal value [64, 69], one finds

$$|a(E)|^2 = \frac{\Theta(E^2 - 4J_q^2)}{1 - \left.\frac{dF(E)}{dE}\right|_{E=E_b}} \delta_{E,E_b} + \frac{\Theta(4J_q^2 - E^2)}{V^2(E)\rho^2(E)\Theta(4J_q^2 - E^2)[\pi^2 + z^2(E)]}$$

$$= \frac{\Theta(E^2 - 4J_q^2)}{1 - \left.\frac{dF(E)}{dE}\right|_{E=E_b}} \delta_{E,E_b} + \frac{\Theta(4J_q^2 - E^2)}{V^2(E)\rho^2(E)\left[\pi^2\Theta(4J_q^2 - E^2) + z^2(E)\Theta(4J_q^2 - E^2)\right]}$$

$$= \frac{\Theta(E^2 - 4J_q^2)}{1 - \left.\frac{dF(E)}{dE}\right|_{E=E_b}} \delta_{E,E_b} + \frac{\Theta(4J_q^2 - E^2)}{V^2(E)\rho^2(E)\pi^2 \left\{ \Theta(4J_q^2 - E^2) + \frac{\left[E - 2\alpha + \sqrt{E^2 - 4J_q^2}\,\Theta(E^2 - 4J_q^2)\right]^2}{4J_q^2 - E^2} \right\}}$$

$$= \frac{\Theta(E^2 - 4J_q^2)}{1 - \left.\frac{dF(E)}{dE}\right|_{E=E_b}} \delta_{E,E_b} + \frac{\Theta(4J_q^2 - E^2)}{\frac{N-1}{2} \left\{ \Theta(4J_q^2 - E^2) + \frac{\left[E - 2\alpha + \sqrt{E^2 - 4J_q^2}\,\Theta(E^2 - 4J_q^2)\right]^2}{4J_q^2 - E^2} \right\}}, \tag{B.8}$$

where we denote

$$F(E) = \text{P.V.} \sum_{k=1}^{N-2} \frac{V^2(k)}{E - E_q^0(k)} = \frac{E}{2} - \frac{\sqrt{E^2 - 4J_q^2}}{2} \Theta(E^2 - 4J_q^2),$$

for brevity. Since

$$\left.\frac{dF(E)}{dE}\right|_{E=E_b=\frac{\alpha^2 + J_q^2}{\alpha}} = \frac{1}{2}\left(1 - \frac{\alpha^2 + J_q^2}{\sqrt{\left(\alpha^2 - J_q^2\right)^2}}\right),$$

one finally obtains the expression for the survival probability by summing the factor $|a(E)|^2 e^{-iEt}$ over the $N-1$ eigenvalues $E$ (the energies in the continuum and the eventual bound state). The Heaviside step functions in the numerators of Eq. (B.8) separates the sum into two contributions depending whether $E^2 \lessgtr 4J_q^2$. This leads to

$$
\begin{aligned}
p_d(t) &= \left| \sum_E |a(E)|^2 e^{-iEt} \right|^2 \\
&= \left| \frac{\alpha^2 - J_q^2}{\alpha^2} e^{-iE_b t} \Theta(\alpha^2 - J_q^2) + \sum_{k=1}^{N-2} \frac{1}{\frac{N-1}{2}\left[1 + \frac{\left(E_q^0(k)-2\alpha\right)^2}{4J_q^2 - E_q^0(k)^2}\right]} e^{-iE_q^0(k)t} \right|^2 \\
&= \left| \frac{\alpha^2 - J_q^2}{\alpha^2} e^{-iE_b t} \Theta(\alpha^2 - J_q^2) + \frac{2}{N-1} \sum_{k=1}^{N-2} \frac{1}{1 + \frac{\left[2J_q \cos\left(\frac{\pi}{N-1}k\right)-2\alpha\right]^2}{4J_q^2 \sin^2\left(\frac{\pi}{N-1}k\right)}} e^{-i2J_q \cos\left(\frac{\pi}{N-1}k\right)t} \right|^2 \\
&\simeq \left| \frac{\alpha^2 - J_q^2}{\alpha^2} e^{-iE_b t} \Theta(\alpha^2 - J_q^2) + \frac{2}{\pi} \int_0^\pi dx \frac{1}{1 + \frac{\left(J_q \cos x - \alpha\right)^2}{J_q^2 \sin^2 x}} e^{-i2J_q t \cos x} \right|^2 \\
&= \left| \frac{\alpha^2 - J_q^2}{\alpha^2} e^{-iE_b t} \Theta(\alpha^2 - J_q^2) + \frac{2J_q^2}{\pi(\alpha^2 + J_q^2)} \int_0^\pi dx \frac{\sin^2 x}{1 - \frac{2\alpha J_q}{\alpha^2 + J_q^2} \cos x} e^{-i2J_q t \cos x} \right|^2, \quad \text{(B.9)}
\end{aligned}
$$

which coincides with Eq. (24) of the main text.

Note that, when $\alpha = 0$ and $J_q \neq 0$, one recovers the simpler case where $H_{\text{eff},q}$ is merely a hopping Hamiltonian. Here, the survival probability reduces to

$$
p_d(t) = \left| \frac{2}{\pi} \int_0^\pi dx \sin^2 x\, e^{-i2J_q t \cos x} \right|^2 = \frac{1}{J_q^2 t^2} \mathcal{J}_1^2(2J_q t), \quad \text{(B.10)}
$$

where $\mathcal{J}_\alpha(x)$ is the Bessel function of the first kind. This result is in agreement with previous works [70, 71].

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
