# Peer review of "Phonon dressing of a facilitated one-dimensional Rydberg lattice gas"

_SciPost Physics, doi:SciPost Phys. Core 5, 041 (2022)_

## Round 2 · Referee Report · Anonymous (Referee 1) · 2021-10-5

Strengths

1) Rigorous and clearly conveyed study of the dynamics of a 1D Rydberg lattice gas under excitation facilitation conditions 2) Analytical and numerical treatment of the hamiltonian of the system, providing a comprehensive discussion of its evolution 3) Explanation through Fano resonance theory of the observed inhibition of the relaxation of an initial state with a single Rydberg excitation

Weaknesses

1) The choice of system to study may be better contextualized within the introduction 2) The experimental considerations sections is very valuable and may further benefit from expanding the discussion towards including current common experimental conditions

Report

My overall view about the manuscript is very positive because it studies a topic relevant for the quantum simulation with Rydberg atoms community. Furthermore, the theoretical analysis is rigorous and clearly conveyed. This manuscript meets the acceptance criteria and it will be suitable for publication in SciPost Physics after a minor amendment. Please, find my list of suggestions in the requested changes section.

Requested changes

1) The introduction would benefit by providing an improved contextualization of the choice of studying a 1D Rydberg lattice gas. Several experiments and theoretical studies have explored the facilitation process in a bulk gas and some have investigated the dynamics when an ordered trapping potential is present. The authors have implicitly covered this distinction through the references that they provided, but it may be helpful for the reader to have a more detailed introduction which would directly explain the choice of focusing this study on a 1D lattice gas system. May the existence of several experiments with reconfigurable tweezer arrays trapping ground state and/or Rydberg atoms provide a complimentary reason for this interest? 2) If possible, it may be helpful for the reader to provide an intuitive motivation of the advantage of using center of mass coordinates. 3) In figure 2 the authors describe several effects that emerge with increasing $k/\omega$ ratio. It may be of interest to note that the lower energy bands symmetry center shifts from $E/\Omega=0$ to negative values for larger $k/\omega$ ratios. 4) It may be beneficial to introduce explicitly the trap frequency $\omega$ at the beginning of section 2. 5) Would it be possible to provide an intuitive description of the origin of the repulsive potential shift $\alpha(k)$ when only one Rydberg excitation is present? 6) To simplify the comparison between the energy scales discussed in Sec 3,4 it would be convenient to write $\Gamma$ with the $2\pi$ factor, as done for the other quantities. 7) It would the interesting to expand the discussion in Sec 3,4 to a regime matching most of the current experiments, for example with $\omega=2\pi 100$ kHz and $\Omega=2\pi 20$ kHz. Would the predicted dynamics be close to the ones presented by the authors or would they change significantly? What would happen if the condition $\omega\gg\Omega$ is not respected?

  • validity: high
  • significance: good
  • originality: high
  • clarity: top
  • formatting: perfect
  • grammar: excellent

Author:  Matteo Magoni  on 2021-10-25  [id 1877]

(in reply to Report 1 on 2021-10-05)
Category:
remark
answer to question

Dear Editor,

We thank you for the correspondence and the Referee for their positive report and their constructive suggestions. For the sake of clarity, we list our responses, with a summary of the changes, in the same numerical order as presented by the Referee. 1) The referee writes:

The introduction would benefit by providing an improved contextualization of the choice of studying a 1D Rydberg lattice gas. Several experiments and theoretical studies have explored the facilitation process in a bulk gas and some have investigated the dynamics when an ordered trapping potential is present. The authors have implicitly covered this distinction through the references that they provided, but it may be helpful for the reader to have a more detailed introduction which would directly explain the choice of focusing this study on a 1D lattice gas system. May the existence of several experiments with reconfigurable tweezer arrays trapping ground state and/or Rydberg atoms provide a complimentary reason for this interest?

Our response:

We have improved the introduction with an explicit reference to the experimental motivation behind our work. We have also rearranged the References accordingly. 2) The referee writes:

If possible, it may be helpful for the reader to provide an intuitive motivation of the advantage of using center of mass coordinates.

Our response:

We have added a sentence motivating the use of the center of mass and relative coordinate. Such coordinates indeed allow to reduce the complex N-body problem to a two-body problem, thanks to the kinetical constraints of the dynamics. 3) The referee writes:

In figure 2 the authors describe several effects that emerge with increasing k/ω ratio. It may be of interest to note that the lower energy bands symmetry center shifts from E/Ω=0 to negative values for larger k/ω ratios.

Our response:

We thank the Referee for putting this aspect under our attention. Accordingly, we have added a sentence in the caption of Figure 2 explaining the reason why the energy levels of the bands decrease as the ratio k/w increases. This is a consequence of the constant term which appears in Eq. (11), which is indeed equal to – 2 k^2/w. This constant, which is neglected in the following computations, is instead included when the numerical diagonalization of Eq. (7) is performed. The presence of this (negative) constant lowers the bands of Figure 2.
4) The referee writes:

It may be beneficial to introduce explicitly the trap frequency ω at the beginning of section 2.

Our response:

We thank the Referee for having pointed out that the phonon frequency was not introduced with the Hamiltonian. Now we have included it with all the other parameters. 5) The referee writes:

Would it be possible to provide an intuitive description of the origin of the repulsive potential shift α(k) when only one Rydberg excitation is present?

Our response:

The repulsive potential shift \alpha(\kappa) stems from the fact that the cluster with only one excitation is the only cluster in which there is no Rydberg-Rydberg interaction. Consequently, since there are no mechanical forces, it is completely decoupled from the phonons. 6) The referee writes:

To simplify the comparison between the energy scales discussed in Sec 3,4 it would be convenient to write Γ with the 2π factor, as done for the other quantities.

Our response:

We thank the Referee to point this out. Now \Gamma is written with the 2 pi factor. 7) The referee writes:

It would the interesting to expand the discussion in Sec 3,4 to a regime matching most of the current experiments, for example with ω=2π100 kHz and Ω=2π20 kHz. Would the predicted dynamics be close to the ones presented by the authors or would they change significantly? What would happen if the condition ω≫Ω is not respected?

Our response:

The Section 3.4 regarding ``Experimental considerations“ was changed in some parts to make it more clear and transparent for the reader. In particular, we have modified the inequalities given by Eq.(15) because \Omega and k can be treated as independent quantities that need to be much larger than \Gamma and much smaller than \omega. Differently to the claim in the previous version of the manuscript the Rabi frequency must in fact not be larger than k. This is due to the fact that the displacement operator defined in Eq. (8), on which we apply perturbation theory, depends on the ratio k/w but not on \Omega. To answer the question, if w = 2 pi 100 kHz, then the spin-phonon coupling evaluates to k = 2 pi 40 kHz and consequently k/w = 0.4, which is approximately the situation depicted in the bottom right plot of Figure 2. However, the Rabi frequency evaluates to Omega = w/8 = 2 pi 12.5 kHz and the condition \Omega >> \Gamma is no more strictly respected. This violation makes a coherent Rydberg cluster dynamics more difficult to achieve. If instead the condition \omega >> \Omega is not met, then the perturbative analysis may not be accurate because bands corresponding to different phonon numbers would overlap. However, in this regime, it is of course possible to perform exact numerical diagonalization.

For the sake of clarity, the changes introduced in the manuscript are highlighted in blue. We hope that the current version of the manuscript, thanks to the improvements suggested by the Referee, is suitable for publication in SciPost Physics. Kind regards, Matteo Magoni, Paolo P. Mazza, Igor Lesanovsky

Anonymous on 2022-01-31  [id 2132]

(in reply to Matteo Magoni on 2021-10-25 [id 1877])
Category:
remark
answer to question

Dear Authors,
Thank you for considering and addressing my comments. My overall view about the manuscript is very positive and it is now suitable for publication in SciPost Physics.
Best regards

Anonymous on 2022-02-11  [id 2186]

(in reply to Anonymous Comment on 2022-01-31 [id 2132])

Dear Referee,

We also thank you sincerely for your fruitful comments that contributed to improve the manuscript.
We look forward to the publication in SciPost Physics.

Kind regards,
Matteo Magoni, Paolo P. Mazza, Igor Lesanovsky

---

## Round 2 · Referee Report · Anonymous (Referee 2) · 2022-7-21

Strengths

  1. Relevance.— The system considered in the manuscript is of immense importance in the field of quantum simulations, especially considering the present-day breakthroughs in experiments with Rydberg atoms.

  2. Rigorous analytical treatment.— The Authors have derived a very simple (but elegant) effective Hamiltonian for the phonon-dressed Rydberg atoms in terms of decoupled tight-binding models under the facilitation constraint using thorough analytical calculations.

Weaknesses

  1. Lack of interesting physical results.— In spite of rigorous analytical derivation of the effective Hamiltonian, the physical consequence that follows (Secs. 4 and 5) are not very surprising and can be predicted beforehand from the form of the effective Hamiltonian (Eq. 11). Since, the effective Hamiltonian is a collection of non-interacting tight-binding models, its physical properties are well-known in the literature.

  2. Lack of generality 1.— The choice of system parameters are very specific as they must follow stringent conditions for the perturbative analytical treatment to be valid. Although the Authors have provided experimental conditions for such parameter choices, such conditions are not easy to meet in present-day experiments (as the Authors also pointed out).

  3. Lack of generality 2.— The inhibition to ballistic growth of the initial cluster configuration is only applicable to the cluster of size 1 (for other sizes, the growth just slows down), that happens due to the energy off-balance for the cluster of size 1 as seen in the tight-binding Hamiltonian of Eq. 11. Such a state has no well-defined thermodynamic limit, and preparation of such a state in experiments would be very hard. Moreover, even a small interaction between this state and others, can destroy such inhibition.

Report

In the manuscript, the Authors have considered Rydberg chains under facilitation constraint coupled to phonon modes, where a single cluster of Rydberg excitations are present. Starting from this very complicated interacting Hamiltonian, they have derived an effective atom-only Hamiltonian using perturbative analysis when energy scales involved are well-separated and follow a specific order in their magnitude. The elegance of their treatment is that starting from a complicated interacting Hamiltonian, they have arrived to a simple effective Hamiltonian that is a decoupled set of non-interacting tight-binding models. To my opinion, this is the main result of the manuscript, presented in Sec. 3, that must deserve a publication in some form.

However, given the stringent criteria of SciPost Physics, I cannot recommend the manuscript for the same. See the ‘weaknesses’ section of the report. Specifically, my main criticism is that the manuscript does not provide any interesting/striking physical results (e.g., in Secs. 4 and 5). As said in the ‘weaknesses’ section, the physical results that are in the manuscript can be guessed by inspecting the structure of the effective Hamiltonian. The lack of generalities is of another concern, as mentioned in the ‘weaknesses’ section.

Therefore, I do not recommend the manuscript for SciPost Physics, but recommend it for SciPost Physics Core instead.

Requested changes

I have minor comments/questions that Authors may consider:

  1. After Eq. 5, the Authors have mentioned that they consider $\kappa > 0$, the repulsive case. But this has an unphysical consequence under facilitation condition that $\Delta < 0$, i.e., excited Rydberg state has lower energy than the ground state. I guess, by considering $\kappa < 0$ will solve this without changing any result, as the results depend on $\kappa^2$.

  2. For Fig. 2, the Authors have put a cut-off of 4 (3 phonon at maximum for each site) in the phonon spaces for Eq. 7. It is not justified or shown that the results are converged with respect to this cut-off for the parameter ranges considered. This cut-off may have an effect in the gaps between the zero-phonon and higher bands.

  3. The Authors may comment on the thermodynamic limit of the band structure (Fig. 2), by considering few different system-sizes.

4.Since $\kappa$ and $V_{NN}$ or $\Delta$ are related, the Authors may show in Sec. 3.4 that the present parameters necessarily imply $\Delta \gg \Omega$ to impose facilitation condition.

  • validity: good
  • significance: ok
  • originality: good
  • clarity: top
  • formatting: excellent
  • grammar: excellent

Author:  Matteo Magoni  on 2022-07-25  [id 2684]

(in reply to Report 2 on 2022-07-21)
Category:
remark
answer to question
reply to objection

Dear Editor,

thank you very much for sending us the second report on our manuscript ‘Phonon dressing of a facilitated one-dimensional Rydberg lattice gas’. We are happy to read that the Referee recommends publication of the paper.

However, we do not quite agree with the “weaknesses” identified by the Referee, especially those regarding the lack of interesting results and the non-definiteness of the thermodynamic limit.

In our view the results presented in Secs. 4-5 can be guessed only once the effective tight-binding Hamiltonian is known, but certainly not from the initial Hamiltonian that is formulated at the start of the paper. We regard this analytical reduction of the full Hamiltonian to a set of effective Hamiltonians of uncoupled phonon-dressed tight-binding models as an interesting achievement on its own.

The concern regarding the thermodynamic limit is also not clear to us. Taking the thermodynamic limit would not change the results of the paper. The state with one excitation would be perfectly defined and would still be energetically off-resonant from the other states because the potential shift defined in Eq. (13) is independent of N. Since the renormalization factor in the hopping constant also does not depend on N, the results of the paper would be the same, with the only changes being a denser q-grid and a continuum of energy levels in the zero-phonon band in Fig. 2. Note, that in the calculations based on Fano-Theory we are taking implicitly the thermodynamic limit, e.g. on page 19.

Furthermore, we do not see why preparing a state with a single Rydberg excitation would be particularly hard, even in the “thermodynamic limit”, i.e., in a long chain. This has been demonstrated already in early experiments, e.g., the ones presented in [Physical Review Letters 118, 063606 (2017)].

Therefore, considering the generally rather positive report of the other Referee and their recommendation for publication in SciPost Physics, we wonder whether our manuscript could be considered suitable for publication in SciPost Physics.

We append a response to the minor suggestions/comments raised by the Referee. We also took the occasion to remove some minor typos from the text.

Sincerely yours, Matteo Magoni and Igor Lesanovsky

The referee writes:

  1. After Eq. 5, the Authors have mentioned that they consider κ>0, the repulsive case. But this has an unphysical consequence under facilitation condition that Δ<0, i.e., excited Rydberg state has lower energy than the ground state. I guess, by considering κ<0 will solve this without changing any result, as the results depend on κ^2.

Our response:

The detuning Δ is defined as the difference between the (bare) atomic transition frequency and the laser frequency. This quantity is hence negative when the laser frequency is larger than the frequency of the atomic transition, which needs to be the case if one wants to overcome the interaction energy. The Referee writes that for negative detuning the Rydberg state has an energy that is lower than the ground state energy. This apparent contradiction is resolved by considering the full energy balance, which also involves the laser field: the energy of the ground state atom and the laser field is simply higher than the energy of the Rydberg state with the laser field containing one photon less.

The referee writes:

2.For Fig. 2, the Authors have put a cut-off of 4 (3 phonon at maximum for each site) in the phonon spaces for Eq. 7. It is not justified or shown that the results are converged with respect to this cut-off for the parameter ranges considered. This cut-off may have an effect in the gaps between the zero-phonon and higher bands.

Our response:

The data shown is converged. Truncating the phonon number per site at 2 leads to results which are virtually indistinguishable. Note, that the purpose of the figure is to compare the analytical perturbative calculation with exact diagonalisation. This is only meaningful in a regime, in which the perturbative results are valid, i.e., when the phonon frequency is large and the mixing between the zero-phonon band and the higher bands is small. Hence, one would expect that in this regime states with large phonon number do not contribute.

The referee writes:

3.The Authors may comment on the thermodynamic limit of the band structure (Fig. 2), by considering few different system-sizes.

Our response:

We are not quite sure what the Referee means. Increasing the system size leads merely to a denser grid in q-space. Hence, in the thermodynamic limit the curves become a continuous function in q. They will look exactly like the red curves in Fig. 2.

The referee writes:

4.Since κ and VNN or Δ are related, the Authors may show in Sec. 3.4 that the present parameters necessarily imply Δ≫Ω to impose facilitation condition.

Our response:

We have added a sentence in Sec. 3.4 that explains how the chosen parameters imply this condition.

Anonymous on 2022-07-26  [id 2690]

(in reply to Matteo Magoni on 2022-07-25 [id 2684])

I thank the Authors for their detailed reply to my report. I believe, there have been a few misunderstandings that may arise from my inadequate choice of wordings. Below, I reply to the Authors’ comments in details.

  1. I agree that the results presented in Sec. 4 and 5, can only be guessed beforehand once the effective Hamiltonian is known, and not from the initial interacting Hamiltonian. In my report, I stressed this point couple of times that the derivation of the simple effective Hamiltonian from the initial complicated Hamiltonian is the main result of the Manuscript, which I find very elegant and interesting, and therefore must deserve a publication in some form on its own. However, as a reader, after going through the derivation of Sec. 3 and the form of the effective Hamiltonian, the results of Sec. 4 and 5 do not stand-out on their own, as they are derived from the effective Hamiltonian itself, and not from the full Hamiltonian (specifically, the inhibition dynamics of Sec. 4). Moreover, there is no comparison presented between the dynamics of the full Hamiltonian and that of the effective one, especially for $\kappa/\omega = 0.5$ (Fig. 3 bottom panel), as this choice of $\kappa/\omega$ is close to the borderline of the validity of the perturbative calculation.

  2. I am still unconvinced about the generality in the choice of system parameters, as they must follow stringent conditions for the perturbative treatment to be valid.

  3. I understand, my choice of the word ‘thermodynamic limit’ have caused some confusions. My concern about the ‘thermodynamic limit’ was not about the Hamiltonians (effective or the original one). I was concerned, as the number of excitations (or energy) in the initial cluster-1 state does not grow linearly with system-size. But, now I understand that this is not an issue, as the extensivity of the initial state is not required for the ballistic growth of the cluster-size, but only required for a thermalizing dynamics (which is beyond the scope of the present manuscript, of course). However, as I said in the report, the initial states having cluster of size 1 are very special. The number of such states only grows linearly with the system-size, while the atom-only Hilbert space dimension grows as $\sim N^2$ without considering the local phonon degrees of freedom. Rest of the states follow ballistic dynamics as seen in the manuscript. Therefore even a small perturbation in the initial state, and/or small couplings to other states (that are ignored in the perturbative expansion) may hinder the inhibition to the ballistic growth, resulting in ballistic (or may be even thermalizing) dynamics. In experiments, getting rid of such small perturbations from the pure cluster-1 state can indeed be very difficult.

  4. In the minor points, the comment about the thermodynamic limit of the band structure was not about that of the effective Hamiltonian, but about the full Hamiltonian of Eq. 7. I should have been more clear. I apologize to the Authors for that. Specifically, I was curious whether the appearance of the bound state in the band structure (of Hamiltonian Eq. 7), as seen in the bottom panels of Fig. 2, survives large size limit or not.

To summarize, I stand by my earlier assessment, as the manuscript, in its present form, in my opinion, does not meet the acceptance criteria of SciPost Physics.

---

## Round 3 · Referee Report · Anonymous (Referee 2) · 2022-7-27

Strengths

Please see the earlier report.

Weaknesses

Please see the earlier report.

Report

I thank the Authors for their detailed reply to my earlier report. I believe, there have been a few misunderstandings that may arise from my inadequate choice of wordings. Below, I reply to the Authors’ comments in details.

  1. I agree that the results presented in Sec. 4 and 5, can only be guessed beforehand once the effective Hamiltonian is known, and not from the initial interacting Hamiltonian. In my earlier report, I stressed this point couple of times that the derivation of the simple effective Hamiltonian from the initial complicated Hamiltonian is the main result of the Manuscript, which I find very elegant and interesting, and therefore must deserve a publication in some form on its own. However, as a reader, after going through the derivation of Sec. 3 and the form of the effective Hamiltonian, the results of Sec. 4 and 5 do not stand-out on their own, as they are derived from the effective Hamiltonian itself, and not from the full Hamiltonian (specifically, the inhibition dynamics of Sec. 4). Moreover, there is no comparison presented between the dynamics of the full Hamiltonian and that of the effective one, especially for $\kappa/\omega = 0.5$ (Fig. 3 bottom panel), as this choice of $\kappa/\omega$ is close to the borderline of the validity of the perturbative calculation.

  2. I am still unconvinced about the generality in the choice of system parameters, as they must follow stringent conditions for the perturbative treatment to be valid.

  3. I understand, my choice of the word ‘thermodynamic limit’ have caused some confusions. My concern about the ‘thermodynamic limit’ was not about the Hamiltonians (effective or the original one). I was concerned, as the number of excitations (or energy) in the initial cluster-1 (single cluster of size 1) state does not grow linearly with system-size. But, now I understand that this is not an issue, as the extensivity of the initial state is not required for the ballistic growth of the cluster-size, but only required for a thermalizing dynamics (which is beyond the scope of the present manuscript, of course). However, as I said in the earlier report, the initial states having cluster of size 1 are very special. The number of such states only grows linearly with the system-size, while the atom-only Hilbert space dimension grows as $\sim N^2$ without considering the local phonon degrees of freedom. Rest of the states follow ballistic dynamics as seen in the manuscript. Therefore even a small perturbation in the initial state, and/or small couplings to other states (that are ignored in the perturbative expansion) may hinder the inhibition to the ballistic growth, resulting in ballistic (or may be even thermalizing) dynamics. In experiments, getting rid of such small perturbations from the pure cluster-1 state can indeed be very difficult.

  4. In the minor points, the comment about the thermodynamic limit of the band structure was not about that of the effective Hamiltonian, but about the full Hamiltonian of Eq. 7. I should have been more clear. I apologize to the Authors for that. Specifically, I was curious whether the appearance of the bound state in the band structure (of Hamiltonian Eq. 7), as seen in the bottom panels of Fig. 2, survives large size limit or not.

To summarize, I stand by my earlier assessment, as the manuscript, in its present form, in my opinion, does not meet the acceptance criteria of SciPost Physics.

  • validity: good
  • significance: ok
  • originality: good
  • clarity: top
  • formatting: excellent
  • grammar: excellent

Author:  Matteo Magoni  on 2022-07-27  [id 2691]

(in reply to Report 1 on 2022-07-27)

We thank the Referee for their detailed reply.
Given their report and their previous recommendation for publication on SciPost Physics Core, we would proceed in resubmitting the manuscript to SciPost Physics Core.

Best regards,
Matteo Magoni and Igor Lesanovsky

---

## Editorial Decision

published